# GAFormer: Enhancing Timeseries Transformers Through Group-Aware Embeddings

**Jingyun Xiao**[*,1]**, Ran Liu**[1]**, Eva L. Dyer**[1,2*]
1 - Machine Learning Center
2 - Department of Biomedical Engineering
Georgia Institute of Technology, Atlanta, GA, 30332

## Abstract

Analyzing multivariate time series is crucial in numerous domains, yet learning robust and generalizable representations within such datasets remains challenging due to complex inter-channel relationships and non-stationary dynamics. In this paper, we introduce a novel approach for learning data-adaptive position embeddings to incorporate learned spatial and temporal structure into transformer architectures. Our framework introduces group tokens and constructs an instance-specific group embedding (GE) layer that assigns input tokens to a select number of learned group tokens, thereby incorporating structural information into the learning process. Building on this, we propose a novel architecture, the Group-Aware Transformer (GAFormer), which integrates both spatial and temporal group embeddings to achieve state-of-the-art performance on various time series classification and regression tasks. Through evaluations on diverse time series datasets, we demonstrate that GE alone can significantly enhance the performance of several backbone models, and that the combination of spatial and temporal group embeddings allows GAFormer to surpass existing baselines. Moreover, our approach effectively discerns latent structures in data without prior knowledge of the spatial ordering of channels, leading to a more explainable decomposition of the spatial and temporal structure underlying complex timeseries datasets. Code is available at https://github.com/nerdslab/GAFormer.

## 1 Introduction

Multivariate time series (MTS) arise in a variety of domains, from finance and traffic prediction to healthcare (Ghosh et al., 2009; Tsay, 2013; Che et al., 2018). MTS consist of many *channels* of univariate time series, where each channel has its own temporal dynamics, and many channels interact through latent interactions or dependencies across channels. The temporal dynamics, or *temporal structure* of each channel, together with the relationships across different channels, or the *channel-wise structure*, are both important when analyzing time series data (Zerveas et al., 2021; Zhang & Yan, 2022). Being able to learn shared spatial and temporal structures in multivariate time series data is essential for obtaining robust representations and building inferences in downstream tasks.

Transformers have demonstrated impressive performance when being used to extract representations from MTS (Zerveas et al., 2021; Nie et al., 2022). To learn nonlocal interactions across different tokens in our sequence, position embeddings are critical to encode the relative ordering between channels and over different points in time (Vaswani et al., 2017). However, standard approaches for position embedding (PE) that are used in language and vision are used can be problematic for the following reasons:

- For general timeseries datasets, there is **no predetermined ordering** or "spatial position" for different channels. Thus, unlike in language or vision, it is challenging to use positional embeddings to build inductive bias to understand the relationships across channels (Su et al., 2021).
- The relationships across channels and time segments might be **instance-specific**. For example, when localizing an event in the brain (e.g., seizure) using multiple electrodes spanning different

---

*Contact information: jxiao76@gatech.edu; evadyer@gatech.edu.

regions in the brain, the dynamics would be altered depending on the specific brain region where the seizure occurs (Shah et al., 2018). In this case, having a fixed set of positional embeddings may lead to poor generalization.

These characteristics of timeseries make conventional positional embedding methods inadequate and ineffective. Moving forward, it will be necessary to design novel types of embeddings techniques tailored to the spatiotemporal structure of MTS.

In this work, we present a novel framework for learning both channel and temporal structure in time series data and then integrating this information into our tokens through "group embeddings" (Figure 1). Our approach learns a concise set of group-level tokens across the dataset and determines how to adaptively assign them to individual samples based on the similarity between the group embedding and specific sample embeddings. This versatile methodology can be employed across sequences to integrate group-level structures into transformer layers in either space or time.

Building on this idea, we introduce the Group-Aware Transformer (GAFormer). By integrating spatial and temporal group embeddings with a spatiotemporal transformer architecture, GAFormer processes multivariate time series (MTS) data uniquely. It examines interactions across both temporal and spatial dimensions, allowing it to form a more unified understanding of the structure within the data. Additionally, by decomposing our grouping into either the spatial or temporal dimension, we show that this also leads to enhanced interpretability, making GAFormer a more explainable architecture for modeling time series.

We tested our proposed technique on both classification and regression tasks spanning a number of different multivariate time series datasets. Our results suggest that group embeddings can be used to boost performance with various back-

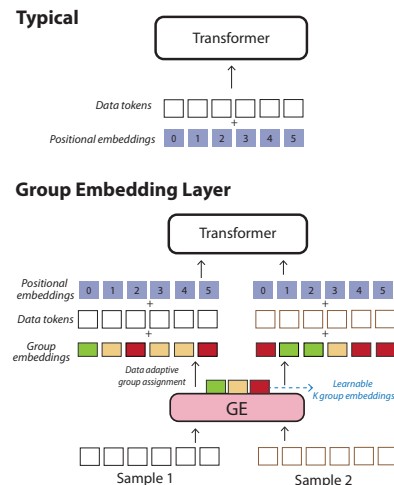

Figure 1: *Group embedding layer*. On top, we show a standard position embedding layer used in transformers. Below, we show our proposed group embedding (GE) layer which learns a set of group tokens and assigns them to different input tokens in data-dependent fashion.

bones, and that by combining both temporal and spatial group embeddings, GAFormer can provide state-of-the-art performance in comparison to previous methods. When we further analyzed the learned group structure, we found that GAFormer can reveal meaningful structure in data without any prior knowledge of the channel or temporal grouping in the data.

The major contributions of our work are as follows:

- We introduce a novel data-adaptive group embedding (GE) technique that can be used to learn both spatial and temporal structure in multivariate time series datasets. GE can be applied flexibly to any transformer encoder, and we show its application in multiple backbones and architectures for incorporating both spatial and temporal group awareness.
- We develop a group-aware transformer, named GAFormer, which provides a robust solution for learning spatial and temporal patterns that leads to improved classification.
- In addition to providing enhanced performance, GAFormer offers meaningful explanations in a variety of different types of time series datasets by revealing both spatial and temporal grouping structures within the data.

## 2 METHOD

In this section, we introduce our approach for building "group embeddings" and then introduce our GAFormer architecture which uses a combination of spatial and temporal group embeddings.

### 2.1 GROUP EMBEDDINGS

Transformers provide a powerful architecture for processing a wide range of data modalities, ranging from text and images, to temporal data. In all of these cases, the raw data must first be "tokenized"

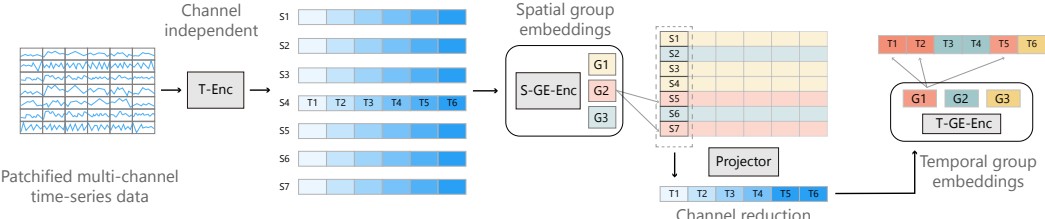

Figure 2: *Combining spatial and temporal group embeddings with a channel-invariant transformer architecture.* To learn both spatial and temporal groupings in multivariate timeseries data, GAFormer starts with a channel-level tokenization of the input before passing the data through a channel-agnostic temporal transformer encoder. In the next stage of processing, we then learn spatial group tokens and embed this information in a spatial group embedding (SGE) encoder; in the last stage, we learn temporal group tokens and embed this information in a temporal group emebdding (TGE) encoder.

to convert it into a sequence of tokens that can be effectively processed by a transformer. After tokenizing our data, we consider a sequence of tokens $X = [\mathbf{x}_1, ..., \mathbf{x}_N] \in \mathbb{R}^{N \times D}$, where $N$ is the total number of tokens in the sequence, and $D$ is the token dimensionality.

Due to the permutation invariant nature of self-attention, position embeddings (PEs) are critical for the success of transformers (Huang et al., 2020; Su et al., 2021). With both learnable and fixed position embeddings (e.g., sin-cos), each token is augmented with a set of non-adaptive embeddings $P = [\mathbf{p}_1, \ldots, \mathbf{p}_N] \in \mathbb{R}^{N \times D}$ to create a new sequence $X_{PE} = [\mathbf{x}_1 + \mathbf{p}_1, \ldots, \mathbf{x}_N + \mathbf{p}_N]$. Thus, in a traditional position encoding scheme, each time a sequence is passed into the model, a token in a specific position in the sequence will be augmented with a fixed embedding associated with that sequence position. We can also write this operation as $X_{PE} \leftarrow X + P$.

In contrast to this fixed scheme for PEs, we propose to build *group embeddings* (GEs) for our input sequence in a data-adaptive manner. To assign the group embeddings to input tokens, we pass the input sequence to an transformer encoder layer $\text{Enc}(\cdot)$ to obtain a sequence $[\text{Enc}(X)_1, \ldots, \text{Enc}(X)_N]$. Each token in this sequence is projected to $K < D$ dimensions through a learned weight matrix $W \in \mathbb{R}^{D \times K}$. After projecting the sequence into a lower-dimensional space, we then apply a softmax function to obtain the weights; this operation will effectively sparsify the coefficients that assign group tokens to input tokens. This group embedding operation $\text{GE}(X)$ can then be written as follows:

$$\text{GE}(X) = \text{SoftMax}(\text{Enc}(X) \cdot W) \cdot G \qquad (1)$$

where $\text{SoftMax}(\cdot)$ represents the softmax function applied to each sequence (column). The group embedding $\text{GE}(X)$ is then added to the input tokens $X$, resulting in:

$$X_{GE} \leftarrow X + \text{GE}(X). \qquad (2)$$

Because we apply the softmax, we can sparsify the assignment weights and thus select a small number of group embeddings to each token. We find that in practice, this assignment is often 1-sparse and tokens in a sequence are each mapped to a single group embedding.

## 2.2 GAFORMER: A GROUP-AWARE SPATIOTEMPORAL TRANSFORMER

Based on the proposed group embedding technique, we propose GAFormer, a spatiotemporal transformer that concurrently extracts both temporal and spatial grouping structures through learning group embeddings in both dimensions (Figure 2). GAFormer is designed to learn representations from multivariate time series $X \in \mathbb{R}^{C \times T}$, where $C$ represents total amount of channels, and $T$ represents total amount of timepoints or patches along the temporal dimension.

**Tokenization Layer:** Following the channel-invariant design of Nie et al. (2022), we first divide the complete temporal sequence from each channel into smaller "patches" (chunks in time), creating a tensor of shape $X \in \mathbb{R}^{C \times P \times L}$, where $P$ denotes the total number of patches, and $L$ represents the number of timepoints in each patch. We then use an encoder $\text{Token}(\cdot)$ to tokenize each patch, forming $Z = \text{Token}(X) \in \mathbb{R}^{C \times P \times D}$. In practice, to both retain the channel-wise separation and also inform the model about the temporal semantics of each channel, we implement $\text{Token}(\cdot)$

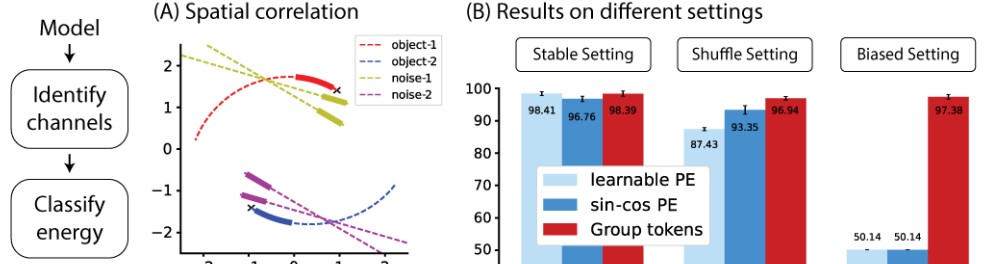

Figure 3: *Synthetic experiments on trajectories from a noisy many-body system.* **(A)** To build the spatial correlation, we use a many-body system with noise, and use the energy of the system to perform classification. Thus, the model first needs to identify relevant objects, and then classify their total energy. **(B)** We compare the performance in three settings: (left) where the channel order is fixed in train and test, (middle) where the channel ordering is permuted during training and testing, and (right) where the channel ordering in training is distributed differently than in test. Our results show that group embeddings can provide robust performance in the presence of channel mismatch and distribution shifts.

as a transformer encoder with learned positional embeddings that processes each channel in $X$ independently.

**Spatial (Channel-Wise) Group Embeddings:** To next extract the channel-wise interactions, we then slice the spatiotemporal sequences spatially, building a set of $P$ sequences of length $C$. Let $Z_S$ denote the new set of $P$ spatial sequences. In this case, we can use our group embedding strategy to learn spatial or across-channel structure through a spatial group embedding (SGE) layer; here, we learn a spatial set of group tokens $G_S \in \mathbb{R}^{K_S \times D}$, where $K_S$ is the number of groups and assignments as described in the previous subsection. The spatial operations jointly update the latent embeddings $Z_S$ as below:

$$Z' = \text{Trans-S}(Z_S + \text{SGE}(Z_S)), \tag{3}$$

where $\text{Trans-S}(\cdot)$ to denote a spatial transformer encoder that operates on sequences of tokens that are at the same point in time but vary across their channel dimension. Critically, since each temporal patch of data is an independent sample to the spatial group embedding module $\text{SGE}(\cdot)$, we can extract different spatial groupings for each time period. This gives GAFormer impressive expressiveness, such that it gives customized spatial grouping to each temporal period based on the structure of the complete training dataset.

**Temporal Group Embeddings:** Next we apply a dimension reduction layer $H(\cdot)$ to extract $Z_T = H([Z'_1, ..., Z'_P]) \in \mathbb{R}^{P \times D'}$, where $C$ channels of $D$-dimension tokens are bottlenecked into one token of $D'$-dimension. We pass this temporal sequence through a temporal transformer encoder $\text{Trans-T}(\cdot)$ and apply a temporal group embedding module $\text{TGE}(\cdot)$ to apply temporal grouping structure to the sequence. This results in a final output which can be written as

$$Z^{\text{final}} = \text{Trans-T}(Z_T + \text{TGE}(Z_T)), \tag{4}$$

where GAFormer maintains a temporal set of group tokens $G_T \in \mathbb{R}^{K_T \times D'}$ with $K_T$ groups.

The architecture is trained end-to-end, where the parameters of the tokenization layer, the spatial encoder, the dimension reduction layer, and the temporal encoder are jointly optimized. When performing discriminative tasks, the tokens in $Z^{\text{final}}$ are averaged and fed into a linear classifier or regressor depending on the task.

## 3   RESULTS

### 3.1   AN INTUITIVE EXAMPLE: NOISY MANY-BODY SYSTEMS

We first provide an intuitive example to motivate the benefits of our proposed group embedding. Specifically, we examine a system where the classification can only be performed *after correctly identifying the channel groupings*. In this scenario, we inspect the shortcomings of existing positional embedding strategies and assess the necessity of employing our proposed GE technique.

**Noisy Many-Body Systems:** To dissect the performance of our model, we consider mutivariate time series generated from the trajectories of many-body systems consisting of mutually interacting particles (Greydanus et al., 2019; Liu et al., 2022). In our experiments, the task is to classify the total energy of a many-body system to decide if it belongs to a *high-energy* or a *low-energy* system. However, to make the problem more challenging, we pollute the system with irrelevant objects that are just passing by and not interacting with the system. To solve this task, we assume that the system needs to (i) identify the objects that are within the interactive system, and then (ii) classify the total energy of the system. We denote the resulting system as the noisy many-body system (Figure 3(A)).

**Experiment Setup:** For each noisy many-body system, we first initialize a pair of interacting objects and a pair of non-interacting objects (Liu et al., 2022), and then solve the trajectories with the Explicit Runge-Kutta method (Dormand & Prince, 1980). We sample 20 consecutive observations with a gap of 0.2 within time span [0, 10] for each system, and randomly generate 30,000 trials to provide rich variability of the trajectories. We compute the total energy of each system and keep the top and bottom quartiles for the classification labels. We then perform a 60/40% train/test random split, and used the same generated dataset throughout.

To benchmark the effectiveness of GE, we train the exact same transformer architecture with (i) learnable positional embedding (Gehring et al., 2017; Radford et al., 2018; 2019), (ii) parameter-free sin-cos positional embedding (Vaswani et al., 2017), and (iii) GE. Each architecture contains 4-layer transformer blocks with 4 attention heads and 32 dimensions, where for GE we share the first transformer layer as the $\mathrm{Enc}(\cdot)$ that generates group embedding coefficients to keep the total depth of the transformer consistent. We train all models with learning rate of 0.0001 using the Adam optimizer (Kingma & Ba, 2014), with a batch size of 64 for 200k steps until the model converges. All runs are repeated for five random seeds.

**Results:** We examine the robustness of GE in three settings: (i) *Stable* setting, where the relative position of objects (channels) never shifts; (ii) *Shuffle* setting, where the observed objects could be in any position and are randomized similarly in training and testing; (iii) *Biased* setting, where the observed objects have different positions that are randomly sampled from non-overlapping sets for the training and testing data.

As shown in Figure 3(B), the synthetic results demonstrate that GE is necessary for the successful modeling of MTS with varying spatial structures. When the channel structure is known and fixed (*Stable*), GE performs similar as the other two position embedding techniques. However, when the channel structure is randomized and thus exhibits rich variability (*Shuffle*), GE shows impressive improvement over learnable position embedding ($\approx\uparrow 10\%$) albeit having similar amount of parameters, and outperforms sin-cos position embedding by a large margin of $\approx\uparrow 4\%$. Finally, when the spatial structure distribution differs from the train and test (*Biased*), GE becomes the only embedding method that gives robust performance ($\approx 97\%$), while the other two baseline methods fail to learn ($\approx 50\%$). The results on synthetic experiments demonstrate that GE is a necessary component for effective learning on datasets where the token-wise structure is unknown or biased.

## 3.2 Time-series classification tasks

In this section, we aim to validate: (i) The effectiveness of GE when it is added into other architectures; (ii) The effectiveness of GAFormer when it is used in multivariate time-series.

**Univariate and Multivariate Datasets:** We validate the effectiveness of GE on both the univariate datasets and the muiltivariate datasets; and further examine the performance of GAFormer on muiltivariate datasets. Both datasets are selected from the UEA Time Series Classification benchmark (Bagnall et al., 2018), where univariate time-series datasets contain InlineSkate (7 classes) (Mörchen, 2006), Earthquakes (2 classes), Adiac (37 classes) (Jalba et al., 2004); While the multivariate time-series datasets contain MotorImagery

Table 1: *Classification performance on univariate time-series datasets. We bold the best model and underline the second best.*

|  | InlineSkate | Earthquakes | Adiac |
|---|---|---|---|
| GRU(Dey & Salem, 2017) | 28.00 | 74.82 | 37.08 |
| TCN(Lea et al., 2017) | 22.55 | 74.28 | 58.06 |
| MVTS(Zerveas et al., 2021) | 22.18 | 74.82 | 57.54 |
| MVTS + TGE | **34.73** | 76.26 | 61.64 |
| $\Delta$ | ↑12.55 | ↑1.44 | ↑4.10 |
| AutoTrans(Ren et al., 2022) | 33.09 | 75.54 | 67.02 |
| AutoTrans + TGE | **34.73** | **76.98** | **75.45** |
| $\Delta$ | ↑1.64 | ↑1.44 | ↑8.43 |

Table 2: *Classification performance on multivariate time-series from the UEA benchmark.* Results obtained through our Group-Embedding (GE) approach are highlighted below each architecture in terms of change in accuracy $\Delta$. The top performing model is boldface and second model is underlined.

| | SelfRegSCP2 (c=7) | FaceDetect (c=144) | Ethanol (c=3) | MotorImagery (c=64) | **Avg.** |
|---|---|---|---|---|---|
| NN | 48.30 | 51.90 | 29.30 | 51.0 | 45.13 |
| $DTW_I$ | 53.30 | 51.30 | 30.40 | 39.0 | 43.5 |
| $DTW_D$ | 53.90 | 52.90 | 32.30 | 50.0 | 47.28 |
| GRU(Dey & Salem, 2017) | 51.11 | 56.56 | 34.60 | 51.0 | 48.32 |
| TCN(Lea et al., 2017) | 53.89 | 66.60 | 30.04 | 50.0 | 50.13 |
| MVTS(Zerveas et al., 2021) | 51.11 | 55.82 | 25.10 | 50.0 | 45.51 |
| MVTS + TGE | 51.67 | 61.75 | 30.42 | 55.0 | 49.71 |
| $\Delta$ | ↑0.56 | ↑5.93 | ↑5.32 | ↑5.0 | ↑4.20 |
| AutoTrans(Ren et al., 2022) | 44.78 | 65.12 | 27.76 | 53.0 | 47.67 |
| AutoTrans + TGE | 52.78 | **68.05** | 27.00 | 56.0 | 50.96 |
| $\Delta$ | ↑8.00 | ↑2.93 | ↓0.76 | ↑3.0 | ↑3.29 |
| PatchTST(Nie et al., 2022) | 50.56 | 54.99 | 25.86 | 54.0 | 46.35 |
| **GAFormer** | **56.11** | 67.99 | **41.44** | **61.0** | **56.64** |

(64-channel ECoG, 2 classes) (Lal et al., 2004), SelfRegSCP2 (7-channel EEG, 2 classes) (Birbaumer et al., 2001), FaceDetect (144-channel MEG, 2 classes), and Ethanol (3-channel Spectrometer, 4 classes) (Large et al., 2018). The selected datasets present a multifaceted spectrum of challenges inherent to time series analysis.

**Baselines:** For univariate experiments, we incorporate temporal GE (TGE) on top of two transformers for time-series classification: MVTS (Zerveas et al., 2021) and AutoTransformer (Ren et al., 2022). We also selected two non-transformer architectures, a GRU (Dey & Salem, 2017) and a TCN (Lea et al., 2017), for comparison.

For our experiments on multivariate time-series, we implemented a supervised version of PatchTST (Nie et al., 2022), apply multivariate versions of the same baselines used for the univariate experiments, and compare against results for a nearest neighbor classifier (NN) (Peterson, 2009) and two dynamic time warping approaches that either use the same ($DTW_I$) or different ($DTW_D$) warping factors across dimensions, as reported in Bagnall et al. (2018). More details of benchmarks and model implementation are stated in Appendix B.

**Experiment Setup:** For each dataset, we perform an 80/20% train/val split on the original training dataset, and select the best model on the validation set to obtain results on the testing set. We perform consistent evaluation on all experiments, where we train all models with the Adam optimizer with an initial learning rate of 0.0003, a cosine annealing scheduler with warm restarts (Loshchilov & Hutter, 2016), a restarting period of 5 epochs, and the multiplying factor of 2. We set the maximum number of epochs as 300. All models are trained until convergence. We provide more details of the models and the optimization process in Appendix B.

**Univariate Results:** In Table 1, we studied the performance of baseline architectures with and without temporal group embeddings on a number of univariate datasets. In all datasets, we show that when adding the temporal group embedding module (TGE), the performance of baseline transformers can be greatly improved. Especially, in InlineSkate, adding TGE to MVTS gives an impressive ↑12.55% increase in accuracy; while in Adiac, adding TGE to AutoTransformer gives ↑8.43% increase in accuracy. The experimental results demonstrate that adding temporal group embeddings to univariate time-series provides an impressive boost in performance.

**Multivariate Results:** In Table 2, we report the classification performance of GAFormer and the performance gain provided by GE across various multivariate datasets. Similar to the univariate results, we show that integrating the temporal group embeddings TGE into the transformer architectures MVTS and AutoTransformer gives notable improvements in accuracy, and in the case of the SelfRegSCP2 dataset, the classification accuracy gain is ↑8.00%. Furthermore, when combining spatial and temporal grouping structures, we show that GAFormer can obtain further boosts in performance, giving an average classification performance increase of ≈↑6%. Our analysis demonstrates the benefits of group embeddings, highlighting its versatility both on its own and when implemented as an independent architecture.

Table 3: *Performance on neural population decoding including both time-series classification tasks (Mihi-Chewie) and regression tasks (NLB). We bold the best model and underline the second best one.*

| | Classification (Acc) | | | | Regression ($R^2$) | |
| | Chewie-1 | Chewie-2 | Mihi-1 | Mihi-2 | NLB-Maze | NLB-RTT |
|---|---|---|---|---|---|---|
| GRU(Dey & Salem, 2017) | 75.00 | 94.44 | 73.81 | 86.05 | 0.8887 | **0.5951** |
| TCN(Lea et al., 2017) | 78.13 | 91.67 | 90.48 | 81.40 | 0.8946 | 0.5407 |
| NDT(Ye & Pandarinath, 2021) | 81.06 | 88.89 | 88.10 | **90.70** | 0.8708 | 0.4621 |
| EIT(Liu et al., 2022) | 75.00 | 77.78 | 78.57 | 65.91 | 0.8791 | 0.4691 |
| **GAFormer** | **81.25** | **94.44** | **92.86** | 88.37 | **0.9136** | 0.5433 |

## 3.3 CLASSIFICATION AND REGRESSION TASKS ON NEURAL RECORDINGS

Brain-computer interfaces (BCIs) enable the direct translation of neural activity into outputs that can control external devices, bridging the gap between the brain and machines. Crucially, the success of BCIs heavily relies on the accuracy of the neural decoding methods, which benefit from the identification of neuronal function groups. Thus, we examine GAFormer for neural decoding tasks.

**Neural Decoding Datasets:** We systematically evaluate GAFormer across six neural decoding datasets that capture neural population activities from the motor cortex of nonhuman primates engaged in different movement tasks (Pei et al., 2021; Liu et al., 2022). In neural recording datasets, the activities of individual neurons are sorted into distinct channels, and the number of spikes across temporal period (20ms) are counted to produce multivariate time-series.

We first test GAFormer on the Mihi-Chewie reaching dataset for classification (Dyer et al., 2017). The dataset consists of stable behavior-based neural responses from different neuron populations and animals performing the same task (Dyer et al., 2017; Liu et al., 2021), where two rhesus macaques, Chewie and Mihi, were trained to reach one of eight locations. While executing different reaches, neural activity in their primary motor cortex was continuously recorded across the two subjects on two different days, forming a total of four sub-datasets.

We also examine the performance of GAFormer on the Maze and the Random Target Task (RTT) tasks of the Neural Latents Benchmark (NLB) for regression (Pei et al., 2021). The Maze dataset records the neural activity from the dorsal premotor (PMd) and primary motor (M1) cortices, where the objective is to predict the hand movement trajectories of the observed subject. The RTT dataset is a unique self-paced sequential reaching task set amidst random elements of a grid. The dataset contains neural spikes from the primary cortex, which are used to predict the subject's hand position.

**Experimental Setup:** We benchmark GAFormer against existing state-of-the-art transformer-based models for neural data, including NDT (Ye & Pandarinath, 2021) and EIT (Liu et al., 2022), as well as traditional methods GRU and TCN. For all experiments, we use the same GAFormer architecture as stated in Section 3.2, and simply add one linear projection layer to the final embeddings of GAFormer to predict the hand velocities for the regression task with an MSE loss. For all experiments, we train the model for 300 epochs, and optimize the network with an Adam optimizer and a cosine annealing learning rate scheduler, and report the converged accuracy and R2 score on the test set. We report additional details about the hyperparameters in Appendix B.

**Results and Insights:** We show the experimental results as in Table 3. Across the board, we find that GAFormer provides strong performance on all neural decoding tasks, with major improvements over the baseline methods in multiple instances. The good performance happens in both classification tasks and regression tasks, where significant improvements in the Chewie-2 and Mihi-1 datasets of a classification accuracy of 94.44% and 92.86% are observed, surpassing the previous state-of-the-art by $\approx\uparrow 3\%$. For the more complicated regression tasks, GAFormer also gives the new state-of-the-art, demonstrating robust decoding performance decoding neural activities. The neural decoding results demonstrate that GAFormer can consistently outperform the previous state-of-the-arts, suggesting its adaptability and promise for a diverse range of tasks.

Overall, the GAFormer model showcases versatility across datasets, highlighting its potential as a robust tool for neural decoding in BCIs, even when compared against the previous state-of-the-art models. Besides the impressive classification performance, another major advantage of GAFormer is its interpretability, which is demonstrated through the visualizations in Appendix C, where the structural information about neurons organizations as well as temporal stages are generated through group embedding assignments automatically.

### 3.4 ABLATION STUDIES

To understand the contributions of various components in our GAFormer architecture, we conducted a number of ablations to the model. These experiments aimed to quantify the impact of temporal group embedding and spatial group embedding within the system.

The first ablation revolved around understanding the performance of GAFormer without any of the group-aware embeddings (Table 4, Base). By comparing this stripped-down version with the full model, we can gauge the performance gain offered by our group-aware embedding mechanism. We found that training the model without group-aware embeddings resulted in suboptimal performance across all of the UEA datasets that we tested. This demonstrates that our proposed group embedding strategy is integral to harnessing the intricate patterns present in multivariate time series.

The second ablation aimed to dissect the individual impacts of spatial and temporal group embeddings. To do this, we trained two distinct model variants: one exclusively leveraging spatial group embeddings (+SGE) and the other reliant solely on temporal group embeddings (+TGE). Both variants improved over the base model with no group embeddings. How-

Table 4: *Ablations.* Our base channel-invariant architecture (left), with temporal GE only, with spatial GE only, and using GAFormer to extract both spatial and temporal GE.

|  | Base | +TGE | +SGE | Ours |
|---|---|---|---|---|
| SelfRegSCP2 | 52.78 | 53.33 | 55.00 | **56.11** |
| FaceDetect | 53.80 | 56.83 | 61.55 | **67.99** |
| Ethanol | 25.10 | 30.04 | 37.64 | **41.44** |
| MotorImagery | 50.0 | 54.0 | 58.0 | **61.0** |

ever, neither reached the performance of the combined spatial and temporal model, indicating that the joint utility of spatial and temporal tokens plays a crucial role in the learning process.

### 3.5 VISUALIZATIONS OF THE GROUP ASSIGNMENTS

A key advantage of our group-aware architecture is that, once trained, we can examine the spatial and temporal group embedding assignments that are learned by the model. Thus, we studied the spatial and temporal embeddings learned by GAFormer in the MotorImagery ECoG dataset (Figure 4) and the NLB RTT dataset (Appendix C.1). In both cases, we found clear temporal grouping structure that appeared to segment the data in a data-adaptive manner, with the temporal grouping structure of NLB RTT aligning well with the intrinsic structure of the movement task (Pei et al., 2021). In the case of the MotorImagery dataset, we also found distinguishable grouping structures along the channel (spatial) dimension. In this case, the imagined movements of fingers have huge variation (Figure 4 left), while the imagined movements of the tongue exhibit similar grouping structures across trials (Figure 4 right). The distinction of grouping structures across the different classes seem to align with the temporal complexity of the task, as the movement of the finger is originally richer than the movement of tongue.

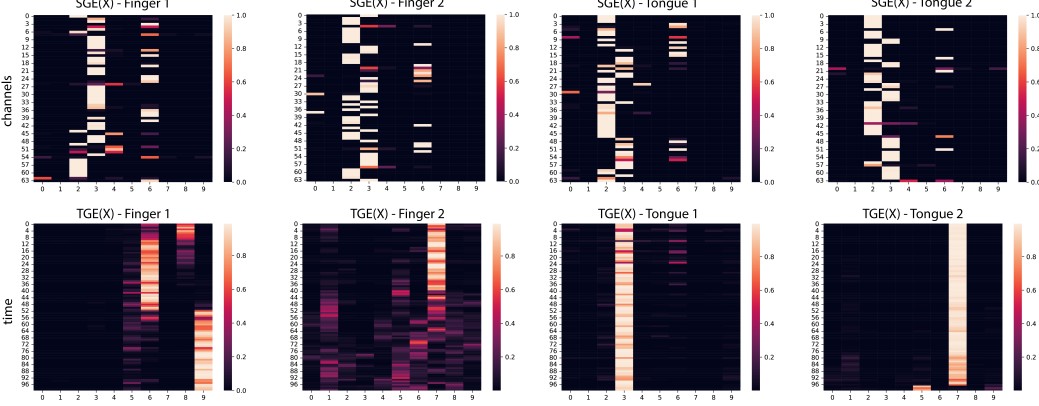

Figure 4: Spatial (top) and temporal (bottom) groupings for the MotorImagery dataset. The two examples on the left and right are from Class 1 (Finger) and Class 2 (Tongue), respectively.

## 4 RELATED WORK

**Position Embedding:** Position embedding plays a critical role in the transformer architecture, whether dealing with textual, images, to temporal data (Huang et al., 2020; Su et al., 2021). Given

the order-agnostic nature of transformers, positional embeddings introduce the essential context needed to build complex relationships across different parts of a sequence. For this reason, numerous studies Li et al. (2021); Zheng et al. (2021); Ke et al. (2020); Wu et al. (2021), have delved into the specifics and refinements of simple positional embeddings. Among existing techniques, our approach shares similarities with the conditional positional embedding that is used in visual representation learning (Chu et al., 2021). However, their method focuses on instance-specific learning of positional embedding without specifically targeting the discovery of grouping structures, which is one unique advantage of our proposed approach. Moreover, it is worth noting that recent works on multivariate time series also explored the use of spatiotemporal embeddings that have been developed for traffic forecasting (Liu et al., 2023) and future location prediction (Lin et al., 2021). Different from them, the robustness of GE relies on the design of the grouping structure, which effectively identifies distinct spatial and temporal structure in MTS datasets.

**Set Discovery and Prediction:** In the domain of computer vision, Locatello et al. (2020) introduced slot attention, which is used for weakly supervised set discovery and prediction problems. Drawing parallels with our proposed methodology, both techniques require the construction of a fixed-size 'group tokens', which are referred to as 'slots' in slot attention. However, our proposed group embedding differs from slot attention and its successive works (Patrick et al., 2021; Siméoni et al., 2021; Ibrahim et al., 2023) due to the unique multivariate nature of timeseries, and the data-adaptive nature of our approach. We hypothesize that the performance gain obtained by our approach is due to the uniqueness of the structure of MTS, which is often neglected in previous works.

**Transformers for Multivariate Timeseries:** Initial approaches to integrating transformers for multivariate time series involved tokenizing short context windows from sequences. These tokenized segments were subsequently embedded using Multi-Layer Perceptrons (MLP) or Temporal Convolutional Networks (TCN) (Zerveas et al., 2021). Despite its foundational nature, this method exhibited limitations in extracting channel-specific features and incorporating them into the training process. Recent methodologies, notably EIT (Liu et al., 2022) and PatchTST (Nie et al., 2022), introduced a channel-independent design, for classification and forecasting tasks, respectively. Building on this work, Wang et al. (2023) introduces a spatiotemporal channel-independent transformer architecture for traffic prediction.

In the realm of forecasting, PatchTST (Nie et al., 2022) has introduced an innovative patch design tailored for time series predictions, maintaining a channel-invariant design. Similarly, CrossFormer (Zhang & Yan, 2022) prioritizes patch designs but with a focus on exploiting inter-channel dependencies. Such methodologies underscore the importance of channel relationships for enhancing forecasting outcomes. Other significant contributions in this domain include works like (Zhou et al., 2021; 2022; Wang et al., 2024), which have expanded the use cases of transformer-based forecasting.

## 5 DISCUSSION

In this work, we introduced a novel framework for building data-adaptive position embeddings to enhance time series transformers. By adaptively assigning group-level tokens to individual samples based on similarity, our method effectively integrates channel-wise and temporal groupings into the transformer architecture. Empirical validations on several time series benchmarks demonstrate how group embeddings can be integrated into a number of existing transformer models. Furthermore, the Group-Aware Transformer (GAFormer) architecture developed in this work offers a robust solution for capturing and leveraging the complex interactions within multivariate time series data, leading to enhanced performance and interpretability without the need for known ordering or spatial structure across channels.

**Limitations and Future Work:** While GAFormer has shown promising results, it partially relies on the channel-independent design as proposed in Nie et al. (2022); Liu et al. (2022). While this approach has many advantages, as discussed in Han et al. (2023), a channel-separable design might not adequately capture the complexities of data with intricate inter-channel dynamics, leading to suboptimal representations in certain datasets. In addition, as the spatial and temporal dimensions grow, more data is needed for the model to effectively extract reliable grouping structures. Moving forward, improving the group embedding module for effective training on high-dimensional MTS data would be an exciting line of research. Additionally, a more in-depth examination of model interpretability, possibly using advanced quantification metrics or visualization tools, can shed light on how the group tokens capture and represent data dynamics.

## ACKNOWLEDGEMENT

This project is supported by NIH award 1R01EB029852-01, NSF award IIS-2039741, NSF award IIS-2212182, NSF CAREER award IIS-2146072, as well as generous gifts from the Alfred Sloan Foundation, the McKnight Foundation, the CIFAR Azrieli Global Scholars Program.

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

## A  DATASETS

### A.1  UEA DATASETS

Table 1: Selected datasets from UEA Multivariate Time Series Classification archive.

| Dataset | Train | Test | Channels | Length | Classes |
|---|---|---|---|---|---|
| MotorImagery | 278 | 100 | 64 | 3000 | 2 |
| SelfRegSCP2 | 200 | 180 | 7 | 1152 | 2 |
| FaceDetection | 5890 | 3524 | 144 | 62 | 2 |
| Ethanol | 261 | 263 | 3 | 1751 | 4 |

### A.2  NEURAL DATASETS

Table 2: Neural datasets.

| Dataset | Train | Test | Units | Length | Classes |
|---|---|---|---|---|---|
| NLB-Maze | 1721 | 574 | 182 | 140 | - |
| NLB-RTT | 810 | 270 | 130 | 120 | - |
| Mihi-Day 1 | 167 | 42 | 162 | 76/84 | 8 |
| Mihi-Day 2 | 172 | 43 | 152 | 73/87 | 8 |
| Chewie-Day 1 | 127 | 32 | 163 | 81 | 8 |
| Chewie-Day 2 | 144 | 36 | 148 | 75/77 | 8 |

## B  EXPERIMENT DETAILS

### B.1  MODEL IMPLEMENTATION AND HYPERPARAMETERS

For the patching tokenization layer of PatchTST and GAFormer, we set the patch window as 10 for SelfRegSCP2, MotorImagery, and Ethanol, set the patch window as 2 for FaceDetec since the data length is short. For all neural datasets (Mihi-Chewie, NLB-Maze, NLB-RTT), we set the patch window as 1 to better evaluate the potential of our model for studying fine-grained dynamics of neural activities. For all experiments and all models including GRU, TCN, MVTS, AutoTrans, the token dimension and embedding size are 256. For Transformer layers, the number of head is set as 16. In GAFormer, we set the depth of spatial/temporal group embedding module and spatial transformer encoder as 3, and the depth for final temporal transformer encoder as 6. For the added TGE module to other baseline models, we kept the depth and dimension consistent with the TGE in GAFormer. For fair comparison, the depth of MVTS, AutoTrans, NDT and EIT are kept the same as the total depths of spatial and temporal Transformer layers in GAFormer. We set the number of groups to be 10 in all group embedding modules. Following Zerveas et al. (2021) and Nie et al. (2022), we adopt batch normalization rather than layer normalization in all Transfomer architectures.

### B.2  TRAINING

We did not use pretraining or data augmentations for all experiments except for the MotorImagery dataset. In the MotorImagery dataset, the number of samples is limited and the length of each trial

| TGE | T-Encoder | Dim | Head | Group K | Patch size |
|---|---|---|---|---|---|
| 3 | 6 | 256 | 8 | 10 | 10 |

Table 3: *Hyperparameters used for univariate datasets.*

| Dataset | Channel-independent Encoder | SGE | S-Encoder | TGE | T-Encoder | Dim | Group K | Patch Size |
|---|---|---|---|---|---|---|---|---|
| FaceDetect | 3 | 3 | 3 | 3 | 6 | 256 | 8 | 10 |
| MotorImagery | 3 | 3 | 3 | 3 | 6 | 256 | 8 | 10 |
| SelfRegSCP2 | 3 | 3 | 3 | 3 | 8 | 256 | 8 | 10 |
| Ethanol | 3 | 3 | 3 | 3 | 8 | 256 | 8 | 10 |
| Mihi-Chewie | 3 | 3 | 3 | 3 | 6 | 256 | 16 | 1 |
| NLB | 3 | 3 | 3 | 3 | 8 | 256 | 8 | 1 |

Table 4: *Hyperparameters used for multivariate datasets.* The first 5 columns represent the depth of each transformer component.

(3000 timepoints) is too long even with patching tokenization. Thus we split each test sample from 3000 timepoints to be 3 samples of 1000 timepoints and formed a larger dataset. In the training stage, we randomly select a segment of 1000 timepoints from each training trial as a strong data augmentation. For models with patching tokenization, we set the batch size as 32. For models with no patching tokenization (MVTS, AutoTrans), we set the batch size as 16 since the number of tokens is large.

## C ADDITIONAL VISUALIZATIONS

### C.1 VISUALIZATIONS ON REAL-WORLD DATASETS

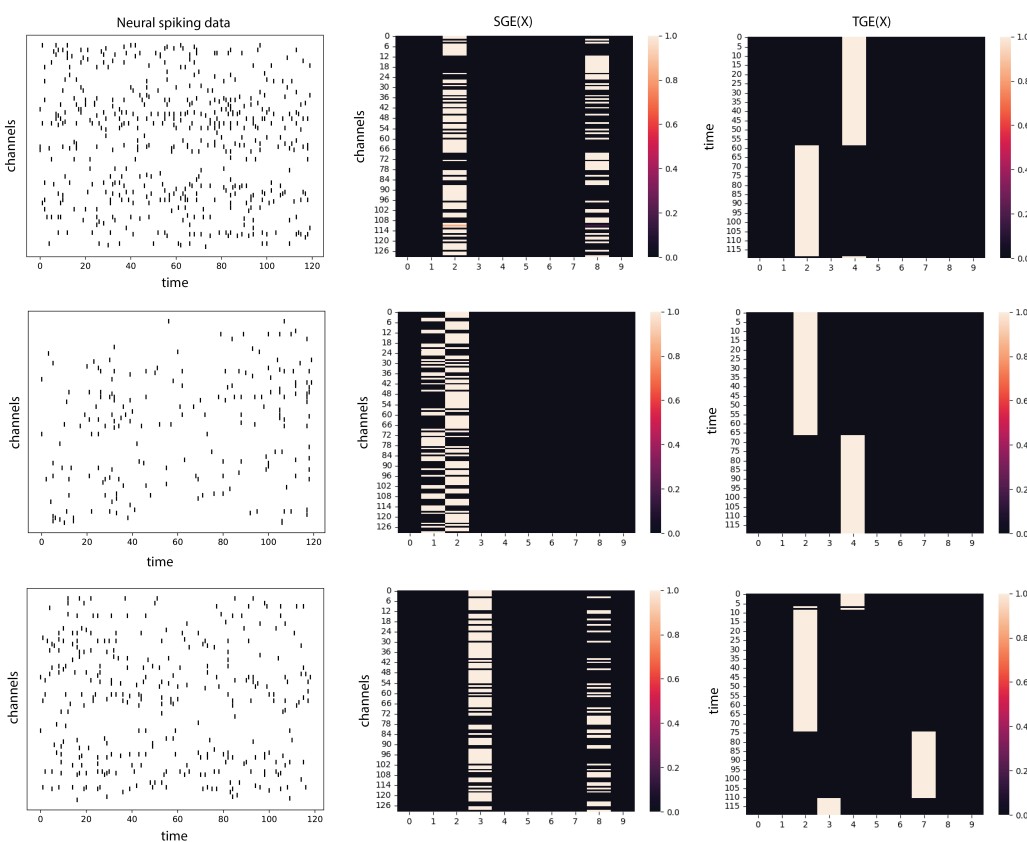

Figure 1: *Visualization of Group Embeddings for Neural Spiking Datasets.* (Left) Spiking data, (Middle) Spatial group embeddings, and (Right) Temporal Group Embeddings for the Neural Latents Benchmark Random Target Task.

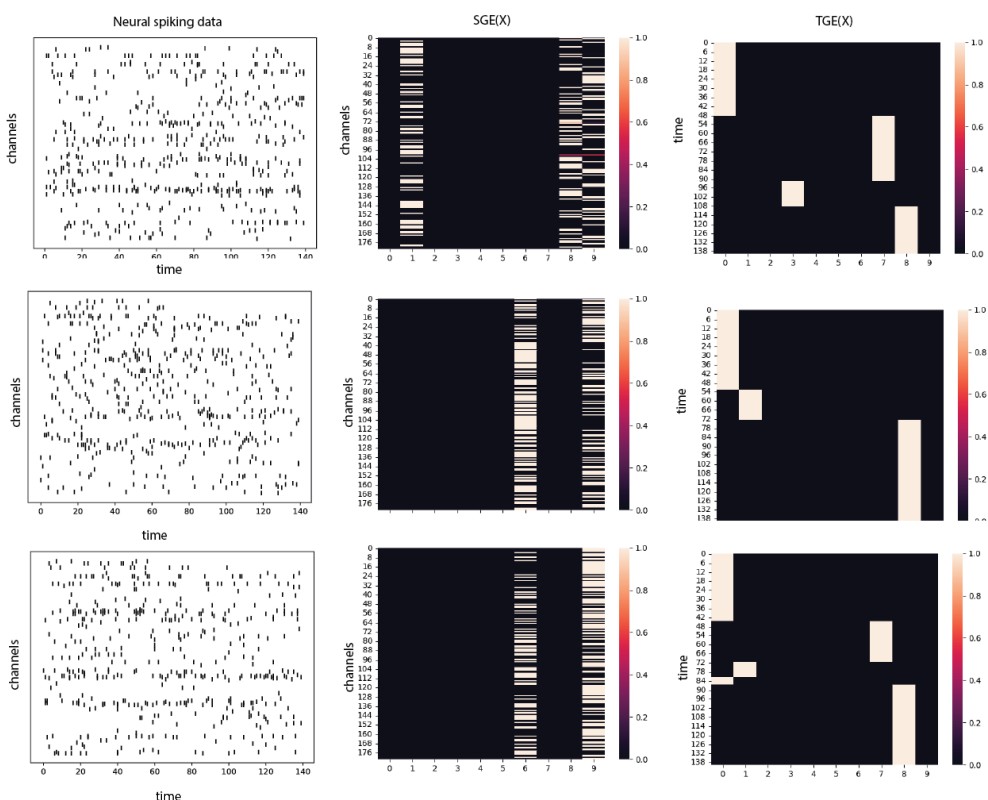

Figure 2: *Visualization of Group Embeddings for Neural Spiking Datasets.* (Left) Spiking data, (Middle) Spatial group embeddings, and (Right) Temporal Group Embeddings for the Neural Latents Maze Task.

As shown on the right in Figure 2, we observed a consistent cutoff in the temporal group assignments, where the initial 50 timepoints were mostly categorized as group 1. We notice that this corresponds to the onset time (at 250ms in each trial, which is binned as the 50th timepoints with bin size of 50ms) of the movements of monkey subjects in the studied **MC_Maze** subdataset (Ye & Pandarinath, 2021). This grouping structure indicates that GE facilitates the encoder's ability to discern nuanced, common structural patterns throughout the dataset.

## C.2 VISUALIZATIONS ON SYNTHETIC DATASETS

We provide additional visualizations of token representations on the synthetic many-body datasets, as shown in Figure 3, Figure 4, and Figure 5.

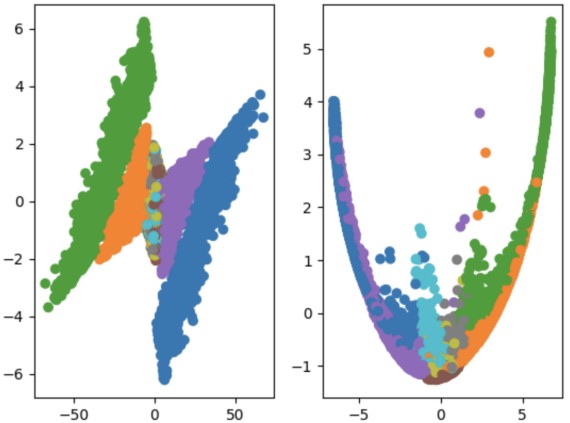

Figure 3: Token embeddings at the input (left) and output (right) layers for a transformer encoder trained with learned position embeddings (PE). Different colors represent the x and y axis of object-1 (green and orange), object-2 (blue and purple), and two random noise objects.

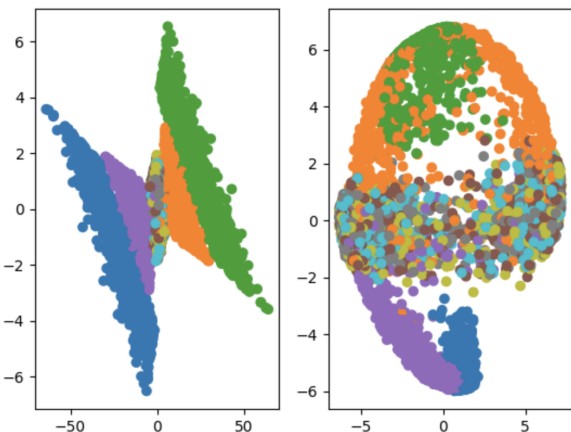

Figure 4: Token embeddings at the input (left) and output (right) layers for a transformer encoder trained with group embeddings (GE). Different colors represent the x and y axis of object-1 (green and orange), object-2 (blue and purple), and two random noise objects.

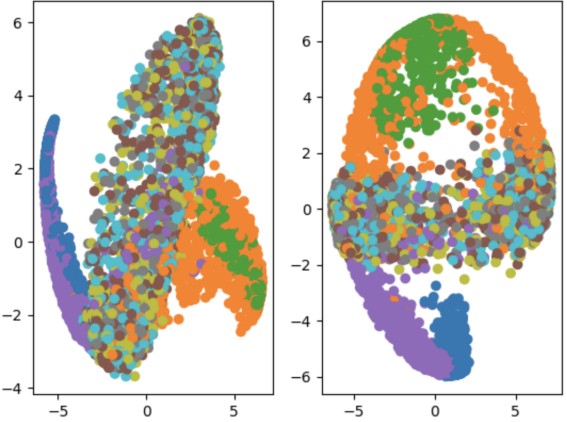

Figure 5: Token embeddings at the output for the same frozen transformer backbone without (left) and with (right) GE. Surprisingly, with the same backbone, applying group embeddings at the input can produce representations that are more separable. Different colors represent the x and y axis of object-1 (green and orange), object-2 (blue and purple), and two random noise objects.

