# OpenReview forum: "GAFormer: Enhancing Timeseries Transformers Through Group-Aware Embeddings"
_ICLR.cc/2024/Conference — ICLR 2024 poster_

### Official Review · Reviewer_MEmY · 2023-10-29

**Soundness:** 3 good
**Presentation:** 3 good
**Contribution:** 3 good
**Rating:** 6
**Confidence:** 2

**Summary:**

In this paper, authors present a novel approach for learning spatiotemporal structure and using it to improve the application of transformers to timeseries datasets. A key aspect of contributions lies in the creation of a specialized group embedding scheme designed specifically for transformer architectures. This scheme enables the adaptive learning of concise grouping tokens, which encompass both channel and temporal dimensions. By incorporating group-aware structural information into the representation space, GAFormer enhances the overall understanding and encoding of data patterns.

**Strengths:**

1.The method proposed by the authors is intuitive and validated on multiple datasets, and the experimental results prove the effectiveness of GAFormer.

2.GAFormer achieves the adaptive discernment of channel-wise and temporal groupings without relying on any predefined structure or additional supervision beyond classification or regression tasks. Through the assignment of group embeddings to tokens, GAFormer enhances the interpretability of the model's representations.

**Weaknesses:**

1.The GAFormer goes through the SGE and then the TGE, what happens if the order of the two modules is reversed? Why is it modeled from the spatial point of view first?

2.Can the author explain Figure 4 again, I don't really understand the interpretability of the authors' proposal.

3.Table 4 shows that SGE is more effective than TGE, can the authors explain why?

**Questions:**

See Weaknesses for details.

---

> ### Author Response · Authors · 2023-11-18
> **Response to Reviewer MEmY**
>
> Dear reviewer MEmY,
>
> We appreciate your time and positive comments on the "effectiveness of GAFormer" and enhancement of the "interpretability of the model’s representations". Based upon your comments, we ran an additional experiment to swap the order of the SGE and TGE operations in GAFormer and report these results below along with additional point-by-point replies to your review.
>
>
> **1. Swapping the order of the SGE and TGE encoders**
>
> Thank you for your insightful question! To better understand the effectiveness of SGE and TGE, we conducted experiments to switch the order of our group embedding layers in GAFormer. In this case, we compare our model with SGE followed by TGE to a model where we apply TGE and then SGE.
>
> | | SGE -> TGE  | TGE -> SGE |
> | -------- | ------- | -------- |
> | MotorImagery | 61.0 | 58.0 |
> | SelfRegSCP2 | 56.11 | 48.33 |
>
> In this experiment, we find that there is a large drop in performance when we switch the order of both operations. This also seems to align with our previous finding inTable 4 where we ablate TGE and SGE and find that SGE provides more significant improvements. Thus, we think that adding SGE at the input layer can have more transformative effects on the model performance, as it learns a discriminative representation for each timepoint by attending to multi-channel relations, which is the basis of the the following TGE block which models inter-timepoint relations. Additionally, we adopt the SGE-TGE architecture to make the model adaptive to different tasks. Some tasks require time-wise representations and some tasks require global representation. We get time-wise representations from the last layer of our model and simply average all timepoints to get global representation.
>
> **2. How group embedding assignments provides insights into the underlying spatial and temporal structure in timeseries datasets**
>
> Thank you for your question, which is an important question in our rebuttal. Please refer to general response 2 for an example of neural spiking data. We show that the learned temporal groups can indicate temporal structures in timeseries data in Appendix Figure 6.
>
> To explain Figure 4, we want to recall that there are 64 channels and 100 timepoints (after the patching layer proposed by [1]) for a single sample in the MotorImagery dataset. For SGE figures (upper row), we have 10 group embeddings and each channel will be assigned with 1 group embedding (the value of Matrix(i, j) indicates the probabilities that channel i is assigned with group j). The visualizations indicate how the 64 channels are grouped within each sample based on the token similarity. Similarly, TGE visulaizations (lower row) indicate how the timepoints are grouped within each sample. And we can also get insights of inter-sample correlations by comparing their visulizations because all samples share the same set of group embeddings.
>
> In contrast to a standard position embedding (PE) which is **fixed** for each datapoint in the dataset, GE is **adaptive to data** and thus for each token and sequence, we build a data-adaptive embedding through a mixture of the learned group tokens. This in turn reveals different grouping assignments for each input token. We find that when we use a GE, we often obtain sparse assignments across the different group tokens, which provides insights into **how different channels or time points are grouped**. In the temporal embeddings, we find contiguous groups that divide the sequence into different phases: the results on neural population dynamics contains integer-valued bins and also results in very sparse and nearly binary group assignments and often correspond to visible transitions in the data (Appendix Figure 6). This point will be elaborated upon in the revision.
>
> **3. Why SGE is more effective than TGE?**
>
> Thank you for your insightful question! We hypothesize that because the temporal structure is naturally preserved across nearby samples in the input sequence, spatial information is more important to harness and learn to inject into the input. Additionally, we hypothesize that the channel-wise relationship is even more critical for classification and regression tasks. This is in contrast to timeseries forecasting tasks [1], where performance remains consistent irrespective of whether the spatial relationship across channels is taken into account. A recent paper [2] also provides extensive discussion on the importance of modeling inter-channel relations.
>
> **Reference**
>
> [1] Nie, Yuqi, et al. "A time series is worth 64 words: Long-term forecasting with transformers." arXiv preprint arXiv:2211.14730 (2022).
>
> [2]Yong Liu, et al. “iTransformer: Inverted Transformers Are Effective for Time Series Forecasting.” arXiv preprint arXiv:2310.06625.

---

> ### Author Response · Authors · 2023-11-21
> **Thanks so much for your constructive feedback**
>
> Dear Reviewer MEmY,
>
> Thanks so much for your comments and feedback. We hope that our additional experiments on reversed SGE, TGE modules, responses on inter-channel relation modeling and explaination of Figure 4 address your concerns. Let us know if there’s anything further that you would like to see or that we can do. Thanks for all your efforts and time throughout the review process!

---

### Official Review · Reviewer_Hkvr · 2023-10-29

**Soundness:** 3 good
**Presentation:** 3 good
**Contribution:** 3 good
**Rating:** 6
**Confidence:** 2

**Summary:**

This paper introduces a novel technique for improving the application of transformers to timeseries datasets by learning spatiotemporal structures. The proposed framework employs group tokens and an instance-specific group embedding layer to integrate structure into the learning process. The newly devised Group-Aware transFormer (GAFormer) demonstrates superior performance across various timeseries tasks, offering enhanced interpretability and the ability to discern latent structures in intricate multivariate datasets.

**Strengths:**

- The idea of using *groupe mbedding*(GE) technique to learn both spatial and temporal structure in multi-variate timeseries datasets is novel.
- The proposed model achieves the state-of-the-art performance on a number of time- series classification and regression tasks

**Weaknesses:**

Though the approach claims to offer "a more interpretable decomposition," this is a subjective claim. The degree of interpretability might vary based on the user or the specific use case.

**Questions:**

- How to tokenise the MTS data?

- What's the different between a text tokeniser and a MTS tokeniser?

- How to determine K?

- What's the effect of different K?

---

> ### Author Response · Authors · 2023-11-18
> **Response to Reviewer Hkvr**
>
> Dear reviewer Hkvr,
>
> We appreciate your time and positive comments to our work. Especially, thank you for identifying our proposed approach as "novel", and recognizing the empirical performance of our model.
>
> **1. How to define interpretability**
>
> Thanks for your insightful question on the definition of interpretability! Please refer to general response 2 for an example of neural spiking data which can provide some intuition of learned groups. We found the learned temporal groups can indicate the temporal structure of the timeseries data.
>
> While we do agree that interpretability is often subjective, the challenges in understanding and extracting the spatiotemporal structure of timeseries data is an outstanding challenge, especially given no pre-assumed underlying structure across channels. We hope our work could further advocate the research along this line, and thus is important to the community.
>
> To address the reviewer’s concern, we attempted to quantify the interpretability through an additional zero-shot fine-grained classification experiment on a synthetic many-body dataset. Specifically, we only train the model to classify the total energy of the system (high energy v.s. low energy, a binary classification task), and later freeze the weight of the transformer encoder. At test time, we extract the token representation of each channel (object), and then train a KNN classifier (k=5, 8-class problem to classify the x, y axis of 4 objects) to cluster and classify the type of each object using the learned token representation. In this case, we can see how GE impacts the learned token representation without providing any information of the type of channel or additional training.
>
> Classification accuracy with positional embedding and group embedding:
>
> | | Positional Embedding  | Group Embedding |
> | -------- | ------- | -------- |
> | Acc | 61.65 | 71.23 |
>
>
> We can see that, with group embeddings, the learned token representation contains more information about the type of the channel in our synthetic setting, demonstrating its interpretability. We believe our architecture can be used to understand the spatiotemporal structure of timeseries datasets, giving the potential to perform zero-shot fine-grained classification.
>
> **2. Details of the Time-Series Tokeniser**
>
> Thank you for your insightful question! We believe that the choice of a “default tokenizer” for time-series is still an active research question, and thus explored two different versions of the tokenizer to make sure that our approach generalizes to both cases. Specifically, we attempted (i) a single-layer MLP to create token representation from a patch of continuous time points of a temporal window from each channel (adopted by PatchTST [1] and GAFormer in our paper); and (ii) a single-layer MLP that creates token representation from all channels of the timeseries at one time point (adopted by MVTS [2], one of the baselines in our paper). The MLP layer is trained end-to-end. Text tokenizer usually builds on a predefined vocabulary and there are discrete word embeddings for each word/token. Different from text tokenizer which process discrete words, time-series tokenizer processes overlapping patches of timeseries, which map continuous timepoints into a higher latent space.
>
>
>
>
> **3. Determine the appropriate size of K**
>
> We conducted additional experiments to examine the robustness of our model as we vary the number of groups (K). The results are provided for MotorImagery and FaceDetect datasets below.
>
> | Values of K | 5  | 10 | 15 | 20 | 25 |
> | -------- | ------- | -------- | ------- | ------- | ------- |
> | MotorImagery | 55.0 | 61.0 | 62.0 | 63.0 | 58.0 |
> | FaceDetect | 67.02 | 67.99 | 67.08 | 68.53 | 68.19 |
>
> We found that our method is relatively stable across different values of K, with only small performance degradation when K is too small or too large. When visualizing the different group assignments it appears that larger values of K will produce a number of tokens with very sparse/few assignments, suggesting that additional unnecessary tokens may not be used and thus similar performance is obtained for the overcomplete case.
>
> We will include these results in an updated version of the paper. Thanks for your suggestion!
>
> **Reference**
>
> [1] Nie, Yuqi, et al. "A time series is worth 64 words: Long-term forecasting with transformers." arXiv preprint arXiv:2211.14730 (2022).
>
> [2] George Zerveas, et al. "A transformer-based framework for multivariate time series representation learning." In Proceedings of the 27th ACM SIGKDD conference on knowledge discovery & data mining, pp.2114–2124 (2021).

---

> ### Author Response · Authors · 2023-11-21
> **Thanks so much for your constructive feedback**
>
> Dear Reviewer Hkvr,
>
> Thanks so much for your comments and feedback. We hope that our additional experiments on different K selection and responses on MTS tokenization address your concerns. Let us know if there’s anything further that you would like to see or that we can do. Thanks for all your efforts and time throughout the review process!

---

### Official Review · Reviewer_cXyc · 2023-11-01

**Soundness:** 3 good
**Presentation:** 3 good
**Contribution:** 3 good
**Rating:** 6
**Confidence:** 4

**Summary:**

The paper proposes a method to learn PEs for transformers in the multivariate temporal data setting. Specifically, the authors propose a Group Embedding (GE) to capture the group aware dynamics along the channels and also the temporal dimension. This Group Embedding is then induced into the transformer architecture called GAFormer to learn the spatial and temporal structure of the multivariate time series datasets. Experimental results on the synthetic dataset show the robustness of the proposed GE over baseline PEs. Moreover, when GE is induced in other transformer architectures for the univariate datasets the performance is enhanced. GAFormer also achieves SoTA performance on the multivariate datasets tested. The authors also claim the interpretability of the proposed method.

**Strengths:**

- Promising PE for multivariate time-series data. The PE show improved performance over the tasks.
- The proposed GAFormer also achieves SoTA performance on the multivariate datasets tested.

**Weaknesses:**

- Since the model is designed to learn the group structure in a data-dependent manner, it may not be able to capture these properties effectively in the low data regime as the baseline PEs would.
- The group structure exhibited in Figure 4 is claimed to provide interpretability of the model. However, these are some groups that have been learnt and the interpretability aspect is not clear as to what it means if some channels or time signals are grouped. This is similar to the groups formed in any attention mechanism and the method doesn’t seem to provide any added explanations or interpretability to the decisions. Thus it may not be right to claim interpretability based on group embedding structure.

**Questions:**

- The authors don’t seem to have mentioned the computational complexity of the method in the paper. From the description, it seems to inherit the quadratic computational complexity of the transformer which is prohibitive for large timesteps and multi chaneled data
- The group structure exhibited in Figure 4 is claimed to provide interpretability of the model. However, these are some groups that have been learnt and the interpretability aspect is not clear as to what it means if some channels or time signals are grouped. Further analysis and explanation by the authors may help.
- As the authors have mentioned other mechanisms (spot attention etc.) to induce group structure, I would expect some empirical study to compare the proposed method with these PEs. Specifically, is the proposed method a unique way to induce PEs for multivariate time series forecasting or could other techniques provide similar or better performance?

---

> ### Author Response · Authors · 2023-11-18
> **Response to Reviewer cXyc**
>
> Dear reviewer cXyc,
>
> We appreciate your time and feedback on our work. Specifically, thank you for commenting on our approach as "promising", and recognizing the improved performance. We provide point-by-point responses to your comments below.
>
> **1. Sensitivity to amount of training data**
>
> Our results are provided for datasets across a wide range of sizes, lengths, and numbers of channels. In all of these cases, we find improvements over standard positional embedding (PE) which we think highlights the flexibility of the approach and its ability to work well even in limited data regimes. For example, the MotorImagery dataset has 278/100 training/test samples, SelfRegSCP2 has 200/180 training/test samples, NLB-RTT has 810/270 training/test samples. Timeseries datasets are usually small compared to the large-scale  text or image datasets.  Exploring the sample efficiency of GE vs PE would be an interesting line of future work.
>
> **2. Interpretability**
>
> We thank you for your comments and agree that our argument for GE providing enhanced interpretability could be explained more clearly. Please refer to general response 2 for an example of neural spiking data. We show that the learned temporal groups can indicate temporal structures in timeseries data in Appendix Figure 6.
>
> In particular, we hope to better highlight the ability of the model to capture both temporal structure and spatial structure in the data. Our approach can do this in two ways: (i) through the choice of assignments to group tokens, and (ii) by improving the representations learned by the model to enhance interpretability or explainability in other ways.
>
> In our extended supplemental figures, we now include more examples and visualizations of the time-series data along with the group assignment coefficients. In these examples, we can see how the switches in dynamics across the time-series are also captured in the group assignment matrix.
>
> **3. Transformer for large timesteps and multi channel data**
>
> Thanks for your question. While our proposed approach does inherit the quadratic computational complexity of the transformer, we would like to point out there are other potential (and maybe better) methods to build GE and learn the group assignment, e.g., CNN, Linear layer, which can be more computing-efficient than transformer. We adopt transformer to build GE because we want to keep the model architectures concise and consistent. The main point of this paper is that group embeddings are effective for enhancing timeseries transformers. Therefore we didn’t search a lot on the building blocks. Designing efficient GE module remains an open question and will be explored in future works.
>
> In the case of large timesteps and multivariate data, PatchTST [1] introduced the method of patching a block of timesteps into a token. This approach significantly reduces the number of input tokens and concurrently enhances model generalization. Our method incorporates a plug-and-play GE module which is relatively light-weight compared to the foundation transformer backbone where it’s added on. Also, the number of tokens for timeseries data are still much lower than the tokens of LLMs, so we don’t worry about the computability of timeseries transformers.
>
>
> **4. Comparison to traditional PE**
>
> We would like to clarify that our baselines and ablations provide different variants of position embedding both in time and in space. However, comparisons with other methods mentioned like slot attention have not been applied to time series datasets and extending them to work across channels is not a trivial task.  To the best of our knowledge, we are the first to explore the concept of group embedding in time-series where there is no spatial information or known correlations across “nearby channels” and thus there are no other PE baselines to compare against.
>
> **Reference**
>
> [1] Nie, Yuqi, et al. "A time series is worth 64 words: Long-term forecasting with transformers." arXiv preprint arXiv:2211.14730 (2022).

---

> ### Author Response · Authors · 2023-11-21
> **Thanks so much for your constructive feedback**
>
> Dear Reviewer cXyc,
>
> Thanks so much for your comments and feedback. We hope that our updates of appendix and responses on sensitivity of training data, computational complexity and computability for large timesteps/channels address your concerns. Let us know if there’s anything further that you would like to see or that we can do. Thanks for all your efforts and time throughout the review process!

---

> > ### Comment · Reviewer_cXyc · 2023-11-22
> >
> > Thank you for responding to my comments and addressing all of them. After reading the paper and rebuttal once again, I find no need for further clarification and will maintain the current score.

---

### Official Review · Reviewer_q3iU · 2023-11-02

**Soundness:** 3 good
**Presentation:** 3 good
**Contribution:** 2 fair
**Rating:** 6
**Confidence:** 3

**Summary:**

The paper tries to improve the transformers to time-series datasets by learning spatiotemporal structure. Specifically, the authors introduce an instance-specific group embedding layer that assigns input tokens to a small number of group tokens to incorporate structure into learning. Based on the group embedding layer, they introduce the GAFormer to incorporate both spatial and temporal group embeddings.

**Strengths:**

1. The paper studies the multivariate time series which is an interesting and important problem.
2. The paper provides a detailed literature review and
3. The paper discusses its limitations.
4. The paper provides extensive experimental results to demonstrate the effectiveness of the proposed method.

**Weaknesses:**

1, The proposed methodology is not well motivated and built on the existing Transformer. The paper tries to introduce the group embedding layer but does not provide clear motivation for the group embedding, i.e. how to define the "group" and why we need the group embedding.\
2. The explanation of the group embedding is vague. It is not easy to understand what the group embedding stands for.\
3. The paper aims to learn generalizable representation within multivariate datasets. I did not find the related experiments to demonstrate the generalization ability.

**Questions:**

1. The authors claim that the group embedding layer can assign input tokens to a small number of group tokens to incorporate structure into learning. I wonder how the structure information can be incorporated using group embedding.

---

> ### Author Response · Authors · 2023-11-18
> **Response to Reviewer q3iU**
>
> Dear reviewer q3iU,
>
> We appreciate your time and insightful comments on our work. We are pleased that you identified our problem as “interesting and important”, and recognized the “extensive experimental results to demonstrate the effectiveness of the proposed method”. In what follows, we provide a point-by-point response to your questions and concerns.
>
> **1. Additional motivation**
>
> Thank you for raising this point about needing to improve the explanation behind group embeddings (GE). We plan to revise the introduction and methods to provide more intuition and explanation of why group embeddings are needed for multivariate timeseries.
>
> Our approach is centered on developing data-adaptive spatiotemporal embeddings, which augment the conventional positional embeddings typically employed in sequential data processing. This is achieved by creating a compact set of universal embeddings that are dynamically combined, often through a sparse linear composition. This process results in customized positional embeddings, varying across different sequences, allowing the model to apply a "group embedding" either to specific channels or points in time. This can be perceived as a form of soft clustering, where input tokens are aligned with group tokens, leading to a shared or similar learned positional embedding. This methodology enables diverse group assignments across various channels and temporal segments when applied to multivariate timeseries, facilitating efficient grouping or segmentation through our approach.
>
> We believe our results underscore the need for new methods to learn spatiotemporal structures that are distinct for each instance, and can be adapted to produce embeddings that are not tied to their absolute position in the sequence. This is particularly pertinent in multivariate timeseries data, where the significance of channels may fluctuate between different classes and across training and testing datasets. Consequently, we believe that both our group embedding approach and the GAFormer architecture present significant advancements, particularly for the timeseries analysis community, where standard Positional Embeddings (PEs) may prove inadequate.
>
> **2. Generalizable representation**
>
> Thanks for your question. By generalization, we were referring to the model’s ability to perform well at test time on a new set of sequences which were not seen by the model in training time. In timeseries, the noise distribution and channel relevance can change abruptly and thus models that aren’t robust to these types of changes in the input may suffer at test time. We will make this clear in the revised paper. Thank you!
>
> **3. Clarification of how we incorporate structure into learning through the GE**
>
> Thank you for your question about “how the structure information can be incorporated using group embedding”. Intuitively, when we solve this assignment problem, multiple channels (SGE) or timepoints (for TGE) will have similar group embeddings and adding this to each input token will induce some underlying shared structure across the tokens with the same group assignments. We will make this more clear in our revision.
> Additionally, we ran two new experiments on our synthetic many-body dataset to provide further insight into how GEs shape learning. Specifically, we train the model to classify the total energy of the system (high energy v.s. low energy, a binary classification task), and then freeze the weight of the transformer encoder. At test time, we extract the token representation of each channel (object), and then train a KNN classifier (k=5, 8-class problem to classify the x, y axis of 4 objects) to classify the type of each object using the learned token representation. In this case, we can see how GE impacts the learned token representation without providing any information of the type of channel or additional training.
>
> Classification accuracy with positional embedding and group embedding:
> | | Positional Embedding  | Group Embedding |
> | -------- | ------- | -------- |
> | Acc | 61.65 | 71.23 |
>
>
> We find that there is a nearly 10\% improvement in our prediction of the channel type when we add GE. These results show that GE shapes the tokens learned at the input to be more aware of structure in the channels, resulting in improved accuracy in the task the model is trained on and in this auxiliary task of interest.

---

> ### Author Response · Authors · 2023-11-21
> **Thanks so much for your constructive feedback**
>
> Dear reviewer q3iU,
>
> Thanks so much for your comments and feedback. We hope that our additional experiments on synthetic many-body dataset and responses on motivation behind GE and generalizable representation address your concerns. Let us know if there’s anything further that you would like to see or that we can do. Thanks for all your efforts and time throughout the review process!

---

> > ### Comment · Reviewer_q3iU · 2023-12-02
> > **Thanks for the response**
> >
> > Thank you for the response and further clarification. After reading the response, I think it addresses most of my concerns and would like to raise the score.

---

### Official Review · Reviewer_Jppq · 2023-11-09

**Soundness:** 3 good
**Presentation:** 3 good
**Contribution:** 3 good
**Rating:** 6
**Confidence:** 2

**Summary:**

This paper introduces a novel transformer architecture called GAFormer, which models the interations of spatial and temporal groups separately. The proposed model demonstrates effectiveness on both synthetic experiments and real-data.

**Strengths:**

1. The paper is clearly written and motivated.
2. The group embedding is novel and very effective as shown by synthetic experiments.

**Weaknesses:**

1. The details of transformer used seems to be missing. see questions
2. Why does this model not compare to (Nie et al.), which seems to be an important related work.

**Questions:**

1. Does this model consisted of 2 layers of transformer? SGE and TGT?

2. What's the specification of these transformers? It should be stated in the main paper.

---

> ### Author Response · Authors · 2023-11-18
> **Response to Reviewer Jppq**
>
> Dear reviewer Jppq,
>
> We appreciate your time and positive comments on our work. Thank you for recognizing our proposed strategy as “novel and very effective”! We provide point-by-point replies to your concerns and questions below.
>
> **1. Details and specifications of our architecture**
>
> Our GAFormer architecture is built based on a cascaded Spatial transformer and a Temporal transformer, with group embedding modules (SGE, TGE) added to both encoders. Specifically, our SGE/TGE module is typically a 3-layer, 8-head, 256-dim transformer with K=10 groups. Please refer to the updated Appendix for more details on the hyperparameters used in our experiments. We will also make sure to include these details in the revised paper. Thanks for your feedback!
>
> **2. Comparison to PatchTST**
>
> While PatchTST has a similar channel separated encoder design, because it is designed for forecasting, we couldn’t directly report numbers from their work on the classification and regression tasks considered here. For our initial submission, we implemented a supervised variant of PatchTST [2] and reported those results in Table 2, where we demonstrate superior performance. Thanks to your comment, we ran additional experiments to also compare with supervised PatchTST for the univariate datasets in Table 1 and the neural datasets in Table 3 (see below). We find that across the board, GAFormer and group embeddings maintain their top performance and the patching-based model without group embeddings. We believe that these results help to further demonstrate the power of our approach.
>
> Additional Baselines for Table 1:
> | | InlineSkate  | Earthquakes | Adiac |
> | -------- | ------- | -------- | ------- |
> | PatchTST | 21.64 | 68.35 | 65.47 |
> | PatchTST + TGE | 21.64 | 69.06 | 69.82 |
> | $\Delta$ | $\uparrow$ 0.00 | $\uparrow$ 0.71 | $\uparrow$ 4.35 |
>
> Additional Baselines for Table 3:
>
> | | C-1  | C-2 | M-1 | M-2 | NLB-Maze | NLB-RTT |
> | -------- | ------- | -------- | ------- | -------- | ------- | -------- |
> | PatchTST (P=5) | 59.38 | 83.33 | 78.57 | 67.44 | 87.57 | 33.40 |
> | PatchTST (P=1) | 78.13 | 86.11 | 88.10 | 72.09 | 88.19 | 43.16 |
> | GAFormer (P=1) | 81.25 | 94.44 | 92.86 | 88.37 | 91.36 | 54.33 |
>
>
> **References**
>
> [1] Liu, Ran, et al. "Seeing the forest and the tree: Building representations of both individual and collective dynamics with transformers." Advances in neural information processing systems 35 (2022): 2377-2391.
>
> [2] Nie, Yuqi, et al. "A time series is worth 64 words: Long-term forecasting with transformers." arXiv preprint arXiv:2211.14730 (2022).

---

> ### Author Response · Authors · 2023-11-21
> **Thanks so much for your constructive feedback**
>
> Thanks so much for your comments and feedback. We hope that our additional experiments on PatchTST comparison and updates of model architectures address your concerns. Let us know if there’s anything further that you would like to see or that we can do. Thanks for all your efforts and time throughout the review process!

---

### Author Response · Authors · 2023-11-18
**General Response**

First, we would like to thank the reviewers for their positive feedback about the work, and for providing validation that we address an “interesting and important problem“ (q3iU), the method is “novel and very effective” (Jppq), “intuitive and validated on multiple datasets” (MEmY) and praise that GAFormer “achieves state-of-the-art performance” (cxYc, Hkvr) on a number of time-series classification and regression tasks; reviewers also noted that the paper is “clearly written and motivated” (Jppq) with a “detailed literature review” (q3iU).
Below, we provide general responses to two key points raised by the reviewers. Point-by-point replies will also be shared with each reviewer.

**1. Intuition and motivation behind our group embedding (GE) approach**

To respond to some of the reviewer's questions about how GE compares with traditional positional embedding (PE) approaches, we think there are two main ways in which GE provides fundamentally different types of information to the transformer at the input layer.


First, PE only captures the positional information that is invariant for each datapoint, while GE is data adaptive and can have different position embeddings for two different sequences. What this means is that if we have an event that occurs at different chunks of time, GE can learn a shift-invariant assignment of the group tokens to the input and deal with this shift. For example, consider two timeseries which are traffic at 8-10AM v.s. traffic at 4-6PM; the traffic jam events may share the same structure in timeseries 1 and timeseries 2, but with fixed PE they have totally different positional embeddings. GE can effectively address this issue.

Second, we can think of GE as performing a global clustering of the **tokens** of the whole dataset and spreading this global information across all samples. We hypothesize that GE adds additional regularization to the learning, which could mitigate overfitting to individual samples and instead promote the learning of features that are broadly applicable across the dataset.

**2. How GE may improve interpretability**

While we do agree with the reviewers that interpretability is both difficult to achieve, define, and is often subjective, we also think that group embeddings provide a unique lens into the structure across and within dimensions of the data that is different from attention scores and the traditional PE. Specifically, we would like to provide the below **additional analysis and experiments** to evidence that GE could improve interpretability.

To assess the impact of the learned GE, we investigated the learned grouping assignment within the neural population decoding dataset **MC_Maze**. We observed a consistent cutoff in temporal group assignments (Appendix Figure 6 RHS), where the initial 50 timepoints were mostly categorized as group 1, corresponding with the onset time (at 250ms in each trial, which should be binned as the 50th timepoints in the setting bin size is 5ms) of the movements of monkey subjects [1]. This grouping indicates that GE facilitates the encoder’s ability to discern nuanced, common structural patterns throughout the dataset.

To further quantify such behaviour and provide further insight into how GEs shape learning, we conducted additional experiments on synthetic many-body dataset. Specifically, we train the model to classify the total energy of the system (high energy v.s. low energy, a binary classification task), and later freeze the weight of the transformer encoder. At test time, we extract the token representation of each channel (object), and then train a KNN classifier (k=5, 8-class problem to classify the x, y axis of 4 objects) to **classify the type of each object** using the learned token representation. In this case, we can see how GE impacts the learned token representation without providing any information of the type of channel during training.

Channel type prediction accuracy with positional embedding and group embedding:
|  | Positional Embedding  | Group Embedding |
|  --| --- | ---- |
| Acc | 61.65 | 71.23 |


We can see that, with group embeddings, the learned token representation contains more information about the type of the channel in our synthetic setting. We believe this experiment helps to underscore how our approach can be used to understand the spatiotemporal structure of timeseries datasets, giving the potential to perform zero-shot fine-grained classification.

We plan to provide additional discussion of the limits of our approach in terms of explainability, and also improve our discussion of the visualization of the group assignments in Figure 4 more clearly.

**Reference**

[1] Pei, Felix, et al. "Neural Latents Benchmark'21: evaluating latent variable models of neural population activity." arXiv preprint arXiv:2109.04463 (2021).

---

### Author Response · Authors · 2023-11-21
**Revised supplementary material uploaded and awaiting reviewers' valuable feedback**

Dear reviewers,

We would like to express our gratitude for your constructive feedback on our submission. In alignment with your valuable suggestions, we have updated the supplemental materials, with all modifications clearly marked in blue to facilitate a swift review. We have also provided additional experimental results and detailed responses to address each reviewer's constructive questions. Here's a brief overview of the key revisions:

 **Appendix**

-We included the hyperparameters used for each dataset, including the details of model architectures.

-We explained that how the learned temporal groups are align with the temporal structure of a neural spiking dataset MC_Maze, showcasing an intuitive case of how our method improves the interpretability of timeseries Transformers.

-We included the visualization of token embedding distribution with and without group embeddings of our synthetic many-body dataset.

**Additional experiments**

-Comparison with PatchTST in Table 1(univariate datasets) and Table 3(neural spiking datasets).

-Results of differen numbers of groups (K) on MotorImagery and FaceDetect datasets.

-Comparison with reversed ordering of SGE, TGE on MotorImagery and SelfRegSCP2 datasets.

-Type classification of learned token representation with or without GE, showcasing how
the structure information can be incorporated when learning with GE.

We would greatly appreciate your confirmation on whether the responses have satisfactorily addressed your concerns. We are ready and willing to discuss any further questions you may have.

Thank you for all your efforts and time throughout the review process!

---

### Meta-Review · Area_Chair_1JVv · 2023-12-09

**Metareview:**

This paper presents a novel group embedding technique for learning spatial and temporal structures in multi-variate time series data. In particular, a group-aware transformer called GAFormer has been proposed for time series classification and regression tasks. Extensive experiments demonstrated that the proposed GAFormer can achieve state-of-the-art performance on various time series tasks. In addition, the experimental results also showed that GAFormer can enhance the interpretability by incorporating the group-aware structural information.

**Justification For Why Not Higher Score:**

While I agree with the reviewers that the method is novel and the evaluation is quite comprehensive, I still think 1) the interpretability part is not that significant and 2) some experimental results (scalability issue, performance on forecasting tasks, etc.) would make the method more impressive. In addition, no reviewers strongly championed this paper (no one gave a score >= 8). Therefore, I prefer not to recommend an Accept with spotlight.

**Justification For Why Not Lower Score:**

All the reviewers reached a consensus, and agreed on both the novelty of the proposed GAFormer model and the comprehensive evaluation. Therefore, I am inclined towards Accept with poster, instead of reject.

---

### Decision · Program_Chairs · 2024-01-16

Accept (poster)